# Moral Distress among Frontline Physicians and Nurses in the Early Phase of COVID-19 Pandemic in Italy

**DOI:** 10.3390/ijerph19159682

**Published:** 2022-08-05

**Authors:** Marina Maffoni, Elena Fiabane, Ilaria Setti, Sara Martelli, Caterina Pistarini, Valentina Sommovigo

**Affiliations:** 1Psychology Unit of Montescano Institute, Istituti Clinici Scientifici Maugeri, IRCCS, 27040 Montescano, Italy; 2Department of Physical and Rehabilitation Medicine of Genova Nervi Institute, Istituti Clinici Scientifici Maugeri, 16167 Genova, Italy; 3Department of Brain and Behavioural Sciences, Unit of Applied Psychology, University of Pavia, 27100 Pavia, Italy; 4Department of Neurorehabilitation of Pavia Institute, Istituti Clinici Scientifici Maugeri, IRCCS, 27100 Pavia, Italy; 5Department of Psychology, Faculty of Medicine and Psychology, Sapienza University of Rome, 00185 Rome, Italy

**Keywords:** moral distress, COVID-19, change in duties, fear of being infected, stress of conscience, healthcare, psychophysical malaise

## Abstract

During the COVID-19 health emergency, healthcare professionals faced several ethical demanding job stressors, becoming at particular risk of moral distress. To date, only a few scales have been developed to evaluate moral distress among frontline professionals working in contact with COVID-19 patients. Moreover, although many healthcare professionals from various disciplines were converted to COVID-19 patient care, no study has yet analyzed whether the resulting change in duties might represent a risk factor for moral distress. Thus, this study aimed to investigate how and when the change in duties during the emergency would be related to healthcare professionals’ psycho-physical malaise. To this aim, a first Italian adaptation of the Stress of Conscience Questionnaire (SCQ) was provided. In total, 272 Italian healthcare professionals participated in this cross-sectional study. Healthcare professionals who had to perform tasks outside their usual clinical duties were more likely to experience moral distress and then psycho-physical malaise. This was particularly likely for those who were extremely concerned about becoming infected with the virus. The results also indicated that the Italian adaptation of the SCQ had a one-factor solution composed of six items. This study provides the first Italian adaptation of SCQ and practical suggestions on how supporting professionals’ well-being during emergencies.

## 1. Introduction

The impact of the ethical component in healthcare decisions is often underestimated. Nevertheless, being prevented from acting according to moral principles can be a source of significant distress and can interfere with professionals’ well-being and the effectiveness and efficiency of their work [1]. More specifically, moral distress arises from situations that endanger an individual’s moral integrity and occurs when, even though individuals know what the ethically desirable action is, they are hindered-unable to implement it due to external obstacles [2]. For example, healthcare professionals may experience feelings of troubled conscience in situations where they feel unable to provide the quality of care they believe is needed [3]. It is not a matter of what is legally acceptable, but rather a subjective perception of contravening one’s professional values and duties [2]. For instance, moral distress can be experienced when the feeling of uncertainty is lamented that it comes from not being scientifically informed about the reasons for therapeutic choices [4]. Moral distress may also be related to personal factors (e.g., managing complex decision-making processes), relationships with patients and caregivers (e.g., communicating breaking bad news), relationships with colleagues (e.g., witnessing patients receiving false hopes), environmental constraints (e.g., lack of resources), and managing ethical dilemmas concerning advance directives [2,5,6].

Previous studies have shown that moral distress is associated with distress, burnout, and sleep problems, ultimately resulting in low job satisfaction, job abandonment, and career change [7,8]. Therefore, accurately measuring this phenomenon and identifying its risk factors in normal and emergency times is of the utmost importance to understanding how to preserve professionals’ well-being and then the healthcare system’s ability to adapt effectively in response to dynamic situations, including health emergencies.

In most health emergencies and during the current COVID-19 outbreak, frontline healthcare providers are confronted with two main ethical issues: whether to react despite the dangers it poses and how to allocate limited, life-saving medical resources [9]. Indeed, in such exceptional situations, healthcare workers may be particularly exposed to priority settings and other ethical dilemmas caused by a scarcity of medical and human resources [10] and an intensive rhythm to carry out new job activities [11,12]. Thus, when resources are limited, clinicians face challenging priority-setting dilemmas due to resource shortages and high-intensity requests when, for example, they have to select patients to admit or treat (i.e., triage decisions) [10,13].

In addition to that, in the current emergency context, particularly during the first COVID-19 wave, the healthcare workforce had to cope with several job-specific ethical demanding stressors [11,13], such as extended work shifts, stringent safety measures, and instructions with few individual rooms for decision. Further job-related stressors included social isolation even stricter than the rest of the population due to the increased risk of contagion and professional demands outside their formal duties [14]. Moreover, especially at the beginning of the pandemic, healthcare professionals were frequently confronted with insufficient personal protective equipment (PPE) [15], the feeling of being inadequately supported, and the absence of certain guidelines for treatment [16]. Among these factors, many studies have identified that healthcare workers were very concerned about being infected by the virus at work and spreading the virus to their families and loved ones [17].

The emergency has also imposed that many professionals were reassigned to work outside their specialty areas with the perception that they did not have adequate training or experience [18]. This aspect has been relatively unexplored, with only a few scientific studies available on this topic in the context of the COVID-19 pandemic. For example, there is evidence that healthcare professionals felt stressed due to changes in their clinical practice and reassignment from their specialty areas to COVID-19 patient care wards during the pandemic [19]. However, to our knowledge, only one study investigated the relationship between changes in usual work tasks during the pandemic and ethical issues [10]. This study by Miljeteig et al., revealed that healthcare workers who were directly involved in the treatment of COVID-19 patients or had been redeployed experienced more priority-setting dilemmas than others, suggesting the need for further research on this relevant risk factor for moral distress.

Beyond this factor, during the COVID-19 pandemic, several job-related stressors (including also those mentioned above) have contributed to increasing the levels of frontline healthcare workers’ moral distress, which was associated with a higher risk of anxiety, depression, post-traumatic stress symptoms, burnout, and psycho-somatic symptoms [16,17,18,20].

Therefore, understanding risk factors for moral distress and its impact on mental health is crucial to inform tailored interventions to support frontline healthcare professionals during current and future health emergencies.

From a research point of view, different instruments have been proposed in the literature to study moral distress [21]. However, moral distress has been mainly investigated through the Moral Distress Scale—MDS, firstly developed by Corley and then revised and validated in many languages and cultures [21]. Although this scale has been used in different healthcare contexts [8,21], its items are particularly focused on palliative care and life-threatening settings. This is the main reason for searching for new scales that can detect moral distress regardless of the specific medical specialty. Additionally, to the best of our knowledge, only a few scales have been specifically developed to assess moral distress related to COVID-19 [22]. In this regard, the Stress of Conscience Questionnaire [3] can be considered a suitable instrument in this context as it has been developed to assess distress resulting from facing ethical demanding situations related to patient care, regardless of the specific medical specialty. The definition of the stress of conscience provided by Glasberg et al., is remarkably similar to that of moral distress [3]. Indeed, the authors claimed that ‘‘stress in health care is affected by moral factors. When people are prevented from doing good, they feel that they have not done what they ought to do, and this gives rise to a troubled conscience’’ [3] (p. 633). In this view, conscience is the “cornerstone of ethics” [23] because when a healthcare professional is aware of what “should be done” but he/she does not have the power or the resources to behave on this awareness, he/she will experience a troubled conscience as a result [23,24,25]. Then, in a broader and more practical sense, we can consider the concept of “moral distress” and “stress of conscience” as synonyms as they refer to a form of existential distress that impacts the conscience of professionals when they are prevented from following their own ethical and moral principles in the provision of care [26]. Moreover, the construct of the stress of conscience has recently been cited when searching for measurements of the subjective experience of moral distress experience [26]. Indeed, feelings of a troubled conscience are the core manifestation of the phenomenon of moral distress [25]. In line with this, the Stress of Conscience Questionnaire [3] specifically developed to assess “stress of conscience” has been described as one of the existing instruments for assessing moral distress [26]. Previous studies in healthcare settings found that this questionnaire is associated with health-related outcomes, such as burnout [27], and may be useful to early predict possible negative workplace outcomes [3,28]. However, as far as we know, the Italian version of this instrument is still lacking, and Italian healthcare organizations may be interested in having an instrument to evaluate their staff perceptions of moral distress during emergencies that can be used for both research and clinical purposes.

An in-depth analysis of moral distress related to COVID-19 challenges can be crucial to detecting ethical conflict situations, identifying the impact of distress indicators, and preventing the long-term consequences for health professionals and, in turn, for the entire healthcare system. Furthermore, measuring the level of moral distress during a sudden health emergency could be essential to evaluate the opportunity to implement prevention interventions in the case of similar situations in the future.

Therefore, this study aimed to investigate how and when the change in duties during the health emergency would be positively related to healthcare professionals’ psycho-physical malaise by analyzing the mediating role of moral distress and the moderating role of concern about being infected with COVID-19 because of one’s job (Figure 1). To achieve this, a first Italian adaptation of the Stress of Conscience Questionnaire (SCQ) was provided by exploring its psychometric properties through principal component analysis and examining its nomological validity. We decided to adopt this measure because recent work by Nillson et al., demonstrated that SCQ was an especially promising measure for assessing moral distress among nurses working in direct contact with COVID-19 patients. The authors found that nurses from three different countries (i.e., Sweden, Denmark, and the Netherlands) reported greater stress of conscience when called to perform unfamiliar tasks compared to those working in their usual wards, indicating that this measure could be particularly suitable to reveal moral distress arising from the change in duties during the pandemic [29]. The authors also found considerable differences across countries and called for more research on this topic in other nations [29]. Then, by providing a first Italian adaptation of the SCQ, we addressed this call while testing how and when the change in duties during the pandemic affected the psycho-physical well-being of healthcare professionals in direct contact with COVID-19 patients. Moreover, according to a recent systematic review, the only Italian instrument to assess moral distress is the translation of the moral distress scale derived from Corley’s moral distress theory [30]. These scales are mainly used in intensive care units, palliative care, and trauma centers [30]. Thus, it would be desirable to provide the scientific community with further instruments to select the most suitable tool for different medical settings. Based on recent literature and item characteristics, the SCQ can be considered a promising instrument to detect moral distress despite the medical specialties considered [29].

Thus, this research contributes to the existing literature on outbreaks and moral distress by identifying this form of stress as a psychological mechanism explaining how the change in duties during the pandemic would be related to psycho-physical malaise among frontline healthcare professionals

## 2. Materials and Methods

### 2.1. Sample

This cross-sectional study was conducted in Italy between April and May 2021.

The research ethics committee of the University of Pavia approved this study. All the procedures performed in this study complied with the ethical standards of the national research committee and with the 1964 Helsinki declaration and its subsequent amendments. Data storage met current Data Protection regulations. All participants provided written informed consent. Participants completed an anonymous self-report survey that was administrated online using a form from a spreadsheet in Google Sheets.

The inclusion criteria were as follows: to be nurses or physicians working in the National Healthcare Sector (NHS) or private contracted healthcare facilities on the Italian territory, and to have been involved in the assistance of COVID-19 patients during the emergency. The exclusion criteria were as follows: to not be nurses or physicians, to not work in contracted private clinics, to be employed in facilities outside Italy, and to have not been involved in the assistance of COVID-19 patients during the pandemic. The questionnaire’s cover sheet informed participants about the study’s goals and ensured both the voluntariness of their participation and the confidentiality of the responses. We distributed the link to submit the survey through social network sites or the company intranet of the hospitals that were involved. Using snowball sampling, starting from the authors’ social networks and personal contacts, research participants were identified who, in turn, recruited more participants directly or through social network groups of healthcare professionals dedicated to sharing experiences and information about clinical practices during the pandemic. Three Chiefs of medicine volunteered to be involved in this study. This allowed us to collect data in three different COVID hospitals in Lombardy, namely the region located in Northern Italy that has been the most affected by the virus. In these hospitals, the Medical Direction (i.e., the board of medical directors that organizes and coordinates physician services and services provided by other professionals within the hospital) approved the study and informed the staff in contact with COVID-19 patients about the research using email via the company intranet. To increase response rates, the Chiefs of medicine sent a follow-up survey email two weeks later to remind staff to participate while highlighting the value of their participation in the realization of the study.

### 2.2. Measurements

The *change in duties* was measured with a single dichotomous (no = 0, yes = 1) item (i.e., Since the pandemic started, have you performed duties other than your usual ones?).

*Psycho-physical malaise* was measured using the twelve-item General Health Questionnaire in its Italian validation [31]. This well-established instrument exists in different versions, but we chose to utilize its twelve-item version because it has consistently shown good statistical properties [32]. This instrument includes three subscales: (a) social dysfunction, which regards difficulties in social performance (six items; e.g., *Have you recently been able to concentrate on what you are doing?*); (b) general dysphoria, which refers to the presence of psycho-somatic symptoms (four items; e.g., *Have you recently lost much sleep over worry?*); (c) loss of confidence, which regards the loss of self-esteem (two items; *Have you recently been thinking of yourself as a worthless person?*). The responses are reported on a four-point Likert scale that measures if and how the person’s mental state is different from his or her usual state (positively worded items: 0 = *better than usual/more so than usual*, 3 = *much less than usual*; negatively worded items: 0 = *not at all*, 3 = *much more than usual*). We decided to use the total GHQ score rather than the three-dimensional model (with items loading solely on each GHQ factor) because the GHQ-12 was originally designed as a one-factor instrument, and numerous scholars advocate the unidimensional utilization of the questionnaire [33]. This instrument provides a total score ranging from 0 to 36 points, where a higher score indicates greater psycho-physical malaise. In the current study, the internal reliability was good (α = 0.82).

*Concern about being infected* with the COV ID-19 virus during one’s job was measured using a single item (i.e., To what extent do you feel concerned about being infected with COVID-19 in the course of your work?) on a four-item Likert scale (1 = not at all, 4 = very much), where higher scores indicate greater concern.

*Control variables*. We controlled for gender (0 = *men*, 1 = *women*), age (in years), job tenure (in years), profession (0 = *nursing profession*, 1 = *medical profession*), having had colleagues who were hospitalized because of the virus (0 = *no*, 1 = *yes*), having lost someone because of the virus (0 = *no*, 1 = *yes*), and most critical period experienced (0 = others, 1 = lockdown). We also controlled for having felt because of one’s job using a single item (i.e., *Have you felt discriminated or treated differently because of your job outside the work context?*) rated on a four-item Likert scale (1 = *never*, 4 = *often*).

### 2.3. Back-Forward Translation of the Stress of Conscience Questionnaire

*Moral distress* was measured using the nine-item Stress of Conscience Questionnaire (SCQ) [3]. This instrument includes items that were developed based on studies on healthcare professionals’ experiences in ethical demanding situations. Each item is divided into two parts: A and B. Part A refers to the frequency with which a certain stressful care situation is experienced. Part A of each item is followed by Part B, which refers to the amount of troubled conscience that the situation described in A generates when it occurs. All items are rated on a six-point Likert scale (from 0 = *never* to 5 = *every day* and from 0 = *no*, *not at all* to 5 = *yes*, *it makes me feel very guilty*/*gives me a very troubled conscience* for Parts A and B, respectively). This instrument gives a total score that reflects total moral distress, which is computed by summing the scores of the products obtained by multiplying Part A (i.e., how often a certain situation occurs) by Part B (i.e., how much troubled conscience the situation evokes when it occurs) for each of the 9 items, giving a total score ranging from 0 to 150.

The SCQ has been designed to address the lack of appropriate instruments measuring the phenomenon related to daily stressful workplace situations in which healthcare personnel perceive that their actions or inactions are in contradiction with their conscience. To this aim, Glasberg et al., developed the SCQ in Sweden on a heterogeneous sample of healthcare professionals (n = 444), not all of whom were working in direct contact with patients [3]. This version listed nine items and had a two-factor structure that explained the 54% of the variance, describing internal demands (i.e., how care is provided; five items, α = 0.74), and external demands and restrictions (i.e., external factors that prevent the provision of optimal care, five items, α = 0.78) [3]. Factors loadings ranged from 0.53 to 0.79. However, there were cross-loadings for three items (items 1, 3, and 8) on both factors and one item (item 1) was included in both factors. The internal consistency of the total scale was α = 0.83 [3]. The stability was investigated with Pearson’s correlations (range: 0.40–0.83 for all items but two with scores of 0.21) [3]. In terms of internal consistency, the SCQ was found to be reliable in Swedish samples of personnel from municipal and community healthcare centers [34,35]. The Swedish SCQ was revalidated by åhlin et al., on healthcare professionals working in direct contact with patients, confirming the presence of two latent factors, although these were different from the original interpretation [36]. The authors did not provide a new interpretation but suggested that the SCQ might be considered unidimensional after the deletion of the item concerning the negative impact of family demands on the work domain (i.e., item 6 “Is your private life ever so demanding that you don’t have the energy to devote yourself to your work as you would”) [36]. Then, the SCQ was validated in Finnish showing a two-factor solution with items that were inconsistent with those of the original validation [37]. Likewise, more recently, the SCQ was adapted to the Dutch context to investigate moral distress among nurses working in direct contact with COVID-19 patients, showing results like those obtained in the Finnish context [29]. Although two factors were identified, most studies have opted to show and interpret the result of the SCQ as a total sum score of all items without using the subscales [34,35]. Additionally, the Australian validation of this instrument showed a single-factor solution composed of nine items with satisfactory factor loadings for a unidimensional structure [28]. Notably, Glasberg et al., and åhlin et al., called for more studies analyzing the SCQ in other clinical contexts, professional groups, and cultural settings [3,36].

Following the World Health Organization’s guidelines for translating questionnaires [38], the forward translation was done by a team composed of three native Italian speakers and one bilingual person. Among these, two were healthcare professionals who were familiar with the vocabulary of the area covered by the instrument and had research skills. Then, a bilingual expert panel, which was composed of three Italian-speaking translators, checked, and discussed the forward translation, identifying appropriate alternatives to inadequate expressions. An independent bilingual translator, who was employed in the healthcare sector and did not participate in the initial translation, translated all items back to English. Finally, both native Italian and English speakers compared the back-translated version with the original version of the scale. There were no relevant differences between the two versions, so this version was approved. Three bilingual physicians were asked to indicate the appropriateness of the translation of the resulting version of the questionnaire during a pre-test, and some amends were made. For each item, the three experts were asked to indicate: (a) to what extent they agreed that the item was representative of the definition of moral distress on a Likert-type five-point scale (1 = *totally disagree*, 5 = *totally agree*); (b) to what extent they believe that the item was essential to the measure of the construct of moral distress on a three-point scale (1 = *useless*, 2 = *useful but not essential*, 3 = *essential*); (c) whether the item was easily understandable (0 = *no*, 1 = *yes*); (d) whether they agreed to keep the item (0 = *no*, 1 = *yes*). An Italian translator supervised the entire translation process.

### 2.4. Statistical Analysis

First, we calculated the content validity indices based on the opinions of the three experts as recommended by Polit et al. [39]. Following Yussof’s recommendations on content validation, we computed the item-level content validity index (I-CVI) by considering the proportion of experts who gave the item a relevance rate above the midpoint of the scale (i.e., agreed item/number of experts) [40]. The Universal agreement (UA) was computed by assigning one to the item that achieved 100% experts in agreement to keep the item. We also computed the scale-level content validity index based on the average method (S-CVI/Ave) by dividing the sum of the I-CVI scores by the number of items and the scale-level content validity index based on the universal agreement method (S-CVI/UA) by dividing the sum of the UA scores by the number of items [40]. I-CVI values equal to 1 were considered acceptable [39]. To confirm the original factor structure of the SCQ, we assessed the descriptive statistics of the items and verified the statistical assumptions (Kaiser-Meyer-Olkin measure of sampling adequacy and Bartlett test groups). Then, we repeated the principal component analysis in SPSS 25 until a factor structure was identified in which all items: (a) loaded onto their primary factor above 0.40, (b) loaded onto alternative factors below 0.30, and (c) had a difference between their primary and alternative factor loadings of at least 0.20 [41]. We checked the communalities and factor loadings for each item, as well as the item-total correlation coefficients. Then, we checked the variance explained by the retained items. As reliability measures, Cronbach’s alpha, McDonald’s omega, Composite Reliability (CR), and Average Variance Extracted (AVE) were calculated in SPSS and JASP, respectively. To confirm the selected factor solution, we conducted a confirmatory factor analysis (CFA) in Mplus using the maximum likelihood (ML) estimation method. The model goodness of fit was verified considering the following indices: the Root Mean Squared Error of Approximation (RMSEA, values below 0.08 and 0.05 indicate, respectively, appropriate and good model fits; [42]), the Standardized root mean square residual (SRMR; values equal to or below 0.08 indicate a good model fit; [42]), the Comparative Fit Index (CFI; values below 0.09 indicate a good model fit; [42]), and the Tucker-Lewis index (TLI; values above 0.09 indicate satisfactory model fit; [42]). Since the chi-square statistic (χ^2^) depends on the sample size, we considered a ratio of the χ^2^ statistic to the respective degrees of freedom (χ^2^/df) below—or equal to—3 as an indicator of superior fit between the expected model and the sample data [43]. Next, we explored the data for descriptive statistics and correlations using SPSS 25. Pearson’s correlation coefficient was deemed appropriate for measuring the strength of the linear relationship between continuous variables. Kendall’s coefficient of rank correlation tau-sub-b was considered appropriate for assessing the association between continuous and ordinal variables. Spearman’s rho correlation coefficient was deemed appropriate for evaluating the correlation between ordinal variables and the correlation between ordinal/continuous and dichotomous nominal variables (in this latter case using the biserial correlation coefficient). The phi coefficient obtained by the contingency table was considered appropriate for calculating the correlation between two discrete nominal variables. To detect possible differences in the study’s continuous variables (i.e., moral distress and psycho-physical malaise) based on socio-demographic variables and COVID-19-related experience variables, we performed independent sample *t*-test analyses and analyses of variance (ANOVAs). Likewise, we explored potential differences in the study’s ordinal variables (i.e., concern about the infection) across groups based on socio-demographic and COVID-related variables using Mann-Whitney tests and Kruskal-Wallis H tests. We also performed a power analysis to determine whether our sample had a sufficient size using G*Power. Then, using the PROCESS macro [44], mediation and moderated mediation models were conducted utilizing bootstrapping tests and a bias-corrected 95% confidence interval (CI) with a resample procedure of 1000 bootstrap samples. These models were performed while controlling for gender, age, job tenure, profession, having had hospitalized colleagues, having been affected by a loss due to COVID-19, having felt discriminated against because of one’s job, and the most critical period experienced. Indirect and conditional effects were considered significant when the *p*-value was less than or equal to 0.05 and zero was not included in the 95% CI. Following Hayes’ recommendations, we used the 16th, 50th, and 84th percentiles of the distribution of the moderator to guarantee that the probed points were within the observed range of the data [45].

## 3. Results

### 3.1. Descriptive Statistics of the Sample

The resulting convenience sample was composed of 272 Italian healthcare professionals. Most of them were women (73.20%) working in the Lombardy Region (90.40%), namely the Italian region that has been the most affected by the virus. Most of the participants worked in the NHS (91.90%), which offers universal access to healthcare. During the pandemic, in Lombardy, the NHS was close to collapse due to the increasing pressure caused by the virus [46]. Before starting to complete the online survey, participants were invited to think about the most critical period they had experienced at work while treating COVID-19 patients and answer the subsequent questions keeping in mind the period selected. Most of the respondents indicated the lockdown (February 2020–May 2020; 73.80%), followed by a later stage of pandemic management (September 2020–February 2021; 19.60%) and the present period (6.60%), as the most critical period experienced at work during the outbreak. Half of the sample worked in the nursing profession (50.00%), while the other half worked in the medical profession (50.00%). Most of the participants were employed in medical wards (21.60%), followed by emergency rooms (18.00%), surgical departments (9.80%), infectious disease wards (9.00%), service of anesthesiology and intensive care (8.60%), COVID-19 units (4.70%), and intensive care units (3.10%). The average age was 45.26 (*SD* = 11.57) with average job tenure of 17.81 years (*SD* = 11.46). Half of the sample reported that, since the pandemic started, they had begun to perform tasks other than their usual ones (50.00%). Of these, more than half referred that this change in duties was on an ongoing basis (54.90%) and did not include additional training (54.60%). All participants reported having colleagues who exhibited symptoms of COVID-19 and, in nearly half of the cases, were hospitalized (52.40%). The respondents believed that they were quite likely to contract the virus in the course of their work (*M* = 7.03, *SD* = 2.33) and felt seldom discriminated against due to their work (*M* = 1.74, *SD* = 0.87). More than half of the sample (53.00%) experienced the loss of a known person (for a more detailed description of the sample see Table 1).

### 3.2. Factor Analysis of the Stress of Conscience Questionnaire

As shown in Table 2, all the nine items from the original scale obtained acceptable I-CVI and UA values and then were kept. Skewness (between −0.16 and 1.41) and kurtosis (between −1.37 and 0.10) indexes showed a normal distribution of the items (i.e., Part A multiplied by Part B according to the original validation [3]). An examination of the Mahalanobis distance scores indicated that there were no multivariate outliers. The Barlett’s Test of Sphericity was significant, and the Kaiser-Meyer-Olkin measure was very satisfactory (0.85), making the factor analysis possible. Following Glasberg and colleagues’ validation [3], we initially hypothesized a two-factor solution, and we conducted an unrotated principal component analysis (PCA) using SPSS in conjunction with a scree plot of eigenvalues. We eliminated items 4, 5, and 6 because they did not meet all the required criteria. This led to selecting, from the original set, six items having a factor loading greater than 0.40 on only one factor (see Table 3). Examination of the scree plot and the total variance explained by the retained factors indicated that the total number of factors to retain was one. Cronbach’s alpha was 0.81, which is similar to that obtained in the original study (i.e., Cronbach’s alpha of the total scale in the original study was 0.83). CR was 0.87 and AVE was 0.52. The average interitem correlation was 0.42. McDonald’s omega was 0.82. The item-total correlation was higher than 0.30 (i.e., 0.42) and all items had a communality higher than 0.20 (i.e., from 0.31 to 0.59). This solution had an eigenvalue of 3.12 and accounted for 52.08% of the explained variance, which is a percentage similar to the original study (i.e., 54.60%) [3]. The corresponding CFA that was conducted using the ML estimation method in Mplus met the criteria for a good fit (χ^2^ = 25.89, df = 9, χ^2^/df = 2.87, CFI = 0.95, TLI = 0.91, RMSEA = 0.08, SRMR = 0.05).

### 3.3. Correlation Analyses

All correlations among variables were in the expected directions (see Table 4). Notably, moral distress was positively associated with psycho-somatic malaise (*r* = 0.60, *p* < 0.01) and concern about being infected (*r_s_* = 0.17, *p* < 0.01), confirming the nomological validity of the first Italian adaptation of the SCQ.

### 3.4. Independent t-Test Analyses and Analyses of Variance

The results of independent *t*-test analyses revealed that there were significant differences based on gender (t_(261)_ = −2.41, *p* < 0.05; see Table 5 and Table 6). Women (*M* = 19.23, *SD* = 7.02) reported higher levels of psycho-physical malaise than men (*M* = 16.97, *SD* = 6.37). Physicians had a greater psycho-physical malaise (*M* = 19.62, *SD* = 6.37) than nurses (*M* = 17.62, *SD* = 7.03). The results of the Mann-Whitney test indicated that there were statistically significant differences based on profession (U_(133,133)_= 6555.50, *z* = −3.17, *p* < 0.01), modalities of change in duties (U_(64,78)_ = 1588.50, *z* = −3.22, *p* < 0.01), received additional training(U_(77,164)_ = 1698.50, *z* = −2.56, *p* < 0.05), and having suffered a loss due to COVID-19 (U_(197,113)_ = 7223.50, *z* = −2.46, *p* < 0.05). Physicians were more concerned about being infected with the virus (*MED* = 3.00, *IQR* = 1.00, *Q*_1_ = 3.00, *Q*_3_ = 3.00, *MR =* 142.96) at work than nurses (*MED* = 3.00, *IQR* = 0.00, *Q*_1_ = 2.00, *Q*_3_ = 3.00, *MR* = 115.63). Healthcare professionals who had to change their duties on an ongoing basis (*MED* = 3.00, *IQR* = 1.00, *Q*_1_ = 2.25, *Q*_3_ = 4.00, *MR =* 56.98) were more concerned about being infected with COVID-19 than those who had to make this change on an occasional basis (*MED* = 3.00, *IQR* = 1.75, *Q*_1_ = 2.00, *Q*_3_ = 3.00, *MR =* 77.60). Healthcare providers who received additional training reported greater concern (*MED* = 3.00, *IQR* = 1.00, *Q*_1_ = 2.00, *Q*_3_ = 4.00, *MR =* 60.85) than those who did not receive it (*MED* = 3.00, *IQR* = 2.00, *Q*_1_ = 2.00, *Q*_3_ = 3.00, *MR =* 77.21). Healthcare professionals who suffered the loss of someone known reported greater concern (*MED* = 3.00, *IQR* = 2.00, *Q*_1_ = 2.00, *Q*_3_ = 4.00, *MR =* 142.16) than those who did not suffer any loss (*MED* = 3.00, *IQR* = 1.00, *Q*_1_ = 2.00, *Q*_3_ = 3.00, *MR* = 120.79).

The results of the ANOVAs and Kruskal-Wallis H tests showed that there were no significant differences based on job tenure and the most critical period experienced. Conversely, the results of the one-way ANOVA indicated that there were significant differences based on age with reference to moral distress (F_(3,268)_ = 5.28, *p* < 0.01). Bonferroni’s post-hoc comparisons indicated that younger healthcare professionals reported higher moral distress levels (*M* = 58.23, *SD* = 38.27) than their older colleagues (*M* = 38.73, *SD* = 35.38). A Kruskal-Wallis H test indicated that there was a statistically significant difference in levels of concern about the infection based on age (χ^2^_(3)_ = 19.92, *p* < 0.001), such that younger healthcare professionals were more concerned about being infected (*MED* = 3.00, *IQR* = 1.00, *Q*_1_ = 3.00, *Q*_3_ = 4.00, *MR* = 152.38) than their middle-aged counterparts (*MED* = 3.00, *IQR* = 2.00, *Q*_1_ = 2.00, *Q*_3_ = 3.00, *MR* = 96.66). The results of the one-way ANOVA showed that there were significant differences based on the perceived quality of the training received in relation to moral distress (F_(2,76)_=6.87, *p* <0.01). The results of Bonferroni’s post-hoc analyses showed that healthcare professionals who considered the additional training received appropriate reported lower moral distress (*M* = 42.35, *SD* = 34.44) than those who considered it inappropriate (*M* = 80.75, *SD* = 29.21). Additionally, A Kruskal-Wallis H test indicated that there was a statistically significant difference in the levels of concern about the infection based on the perceived quality of the training received (χ^2^_(3)_ = 12.56, *p* < 0.01), such that healthcare providers who perceived the training received as inappropriate (*MED* = 4.00, *IQR* = 0.25, *Q*_1_ = 3.75, *Q*_3_ = 4.00, *MR* = 51.39) reported greater concern about being infected than those who evaluated the training received more positively (basic: *MED* = 3.00, *IQR* = 1.75, *Q*_1_ = 2.00, *Q*_3_ = 3.75, *MR* = 33.35; appropriate: *MED* = 3.00, *IQR* = 1.00, *Q*_1_ = 2.00, *Q*_3_ = 3.00, *MR* = 30.47).

### 3.5. Hypotheses Testing

#### 3.5.1. Mediation Analysis

Firstly, power analysis for multiple regression analysis with 10 predictors was conducted in G*Power to determine whether our sample had sufficient size using an alpha of 0.05, a power of 0.95, and a medium effect size (*f*^2^ = 0.15). The results indicated that a sample of at least 173 subjects was required, suggesting that our sample size was adequate. As the collinearity statistics (i.e., Tolerance and VIF) were all within accepted limits (between 0.90 and 1.26), the assumption of multicollinearity was met. Examination of the Mahalanobis distance scores indicated that there was no sign of multivariate outliers. Then, a mediation model was conducted to examine whether moral distress would mediate the link between the change in duties and psycho-physical malaise. The results of this model indicated that the change in duties was positively associated with moral distress (β = 18.66, SE = 4.24, *p* < 0.001, 95% CI [10.31, 27.01]) which, in turn, was positively related to psycho-physical malaise (β = 0.11, SE = 0.01, *p* < 0.001, 95% CI [0.09, 0.13]) and partially mediated the link between the change in duties and malaise (β = 2.03, SE = 0.51, 95% CI [1.02, 3.09]). The indirect effect was positive, suggesting that professionals who had to change their duties reported greater moral distress which, in turn, was positively related to the psycho-physical malaise.

#### 3.5.2. The Moderated Mediation Analysis

Concern about being infected with the virus during one’s job strengthened the positive association between the change in duties during the emergency and total moral distress (β = 9.93, SE = 5.01, *p* < 0.05, 95% CI [0.07, 19.80]; see Table 7 and Figure 2). The moderated mediation effect of the change in duties and the concern about being infected through moral distress on psycho-physical malaise was significant for professionals who had high (β = 30.32, SE = 7.21, *p* > 0.001; 95% CI [16.11, 44.53]) or moderate (β = 20.39, SE = 4.29, *p* < 0.001; 95% CI [11.97, 28.84]) levels of concern, but not for those who had low levels of concern (β = 10.46, SE = 5.91, *p* = 0.08, 95% CI [−1.19, 22.10]). Age (β = −1.14, SE = 0.47, *p* > 0.05; 95% CI [−2.06, −0.21]) and having felt discriminated against (β = 8.48, SE = 2.47, *p* > 0.001; 95% CI [3.61, 13.34]) because of one’s job correlated with moral distress. Age (β = −0.24, SE = 0.08, *p* > 0.01; 95% CI [−0.39, −0.09]) and having more experience (β = 0.25, SE = 0.07, *p* > 0.001; 95% CI [0.11, 0.40]) correlated with psycho-physical malaise. Examination of the interaction plot (see Figure 3) showed that compared to their colleagues with little concern, healthcare providers with great concern about being infected reported a considerable increase in moral distress in the passage from no change in duties to change in duties conditions.

## 4. Discussion

The COVID-19 pandemic has led to growing attention to the psychological health of healthcare professionals. Indeed, since the beginning of the pandemic, healthcare workers had to face critical aspects such as the unexpected and uncontrollably increasing cases and deaths, the severe risk of contagion, deaths, and hospitalizations of their colleagues and loved ones, working on the frontlines with changes in duties, the perceived discrimination against them because of their job [11,12,13,14,15,16,17]. Previous studies have consistently demonstrated that all these risk factors increased the incidence of psycho-physical disturbances among healthcare workers on the frontlines of the fight against COVID-19 [11,12,13,14,15,16,17]. Compared to these studies, empirical research investigating moral distress during the pandemic represents a minority despite the frequent exposure of healthcare professionals to ethical demanding situations from the very beginning of the pandemic [10,13,22].

Within the Italian context, this might also be due to the lack of context-specific instruments. Indeed, to date, several moral distress scales have been specifically designed for healthcare workers (e.g., the Moral Distress Thermometer) [21] and some measures have been targeted for specific groups of healthcare professionals (e.g., The Italian Pediatric Moral Distress Scale-Revised [47]; The Italian version of the Moral Distress Scale for Nursing Students [48]; The Italian Moral Distress Scale-Revised for critical care clinicians [49]). However, there was not yet a reliable instrument to specifically measure morally distressing situations experienced by healthcare professionals during the COVID-19 pandemic. To fill this gap, this study provides a preliminary version of the SCQ in Italian. Unlike the original nine-item scale, which had a two-factor structure, the Italian version of the SCQ was composed of six items and had a one-factor structure. This unidimensional solution was like that obtained from the Australian adaptation of the SCQ [28]. A possible explanation for this difference might be attributable to the dissimilarities existing between the pandemic Italian and the pre-pandemic Swedish healthcare context where the instrument was first developed [3]. Thus, it might be that the three excluded items (i.e., seeing patients insulted, avoiding patients or family members who need help, and having a demanding private life with repercussions on work) could not apply to the work situation that Italian healthcare professionals encountered during the first COVID-19 wave (i.e., most respondents answered the survey thinking about the lockdown period). For instance, it might be unlikely that healthcare providers could escape from patient care or witness situations in which their patients were insulted during the outbreak. Indeed, given the shortages of human resources, all available personnel were required to work extended shifts to manage the high volume of patients, exposing them to direct and prolonged contact with patients. In this context, as demonstrated by numerous studies [50], healthcare providers were exposed more frequently than in the past to verbally aggressive patients who vented on them their negative emotions elicited by the treatment received (e.g., long waiting times, uncertainty about clinical procedures) and the impossibility of seeing their families. In fact, family members were not allowed into hospital facilities due to COVID-19 containment measures, which might have made the item regarding the avoidance of family members who needed support not applicable. Additionally, Italian healthcare workers often had difficulties in taking care of their children and elderly family members because they had to work long shifts and maintain social distancing (sometimes putting themselves in quarantine) to limit the risk of infecting their loved ones [51], experiencing work-family conflict rather than family-work conflict [50].

The correlations were in the hypothesized directions confirming the concurrent validity. Moral distress was positively correlated with psycho-physical malaise and concern about being infected. Additionally, this study found that younger healthcare professionals reported experiencing greater moral distress than their older counterparts. This is consistent with what was found in some previous studies [52] and might be related to the fact that, during the pandemic, Italian healthcare organizations had to recruit inexperienced new graduates due to human resource shortages. Furthermore, the present research revealed that healthcare professionals were less likely to report moral distress when they perceived the additional training received in preparation for their change in duties as appropriate (vs. inappropriate). The importance of the perceived quality of the training received has been underlined by the previous literature on outbreaks, indicating that those who perceive their training as inadequate were more likely to develop burnout, post-traumatic stress symptoms, and longer prolonged perceived risk even after the health emergency was over [53]. Overall, this study provides researchers and practitioners with a reliable instrument to assess the frequency and intensity of different distressing situations (i.e., lack of time, forced to provide care not agreed with, dealing with incompatible demands, not having time for private life, unattainable others’ expectations, lowered aspirations of diligent care) that healthcare professionals can experience during health emergencies.

The main objective of this study was to clarify how and when the change in duties during the pandemic would be positively related to healthcare professionals’ psycho-physical malaise. Our findings showed that moral distress was the psychological mechanism explaining why healthcare professionals who had to perform tasks outside their usual clinical duties were more likely to experience psycho-physical malaise. A possible explanation is that many professionals were assigned to new roles (or wards) and required to perform unaccustomed tasks for which they did not feel adequately prepared. Thus, these duties might be well beyond the scope of their prior practice and outside the usual domain of practice for many professionals, requiring skills in which they were less experienced. For instance, pediatricians working as internal medicine specialists had to rapidly acquire the expertise to attend to COVID-19 patients [54]. These unexpected changes in the sort of work might have posed healthcare professionals in morally distressing situations because they might not have been able to provide patient care in the way they were trained or perform what they believed was ethically correct [7,8,9,10], therefore undermining their psycho-physical health. Thus, our study adds to an increasing body of literature on moral distress during the COVID-19 pandemic [20] by confirming the pivotal role of moral distress in affecting the psycho-physical well-being of healthcare professionals in contact with COVID-19 patients. Additionally, this study enriches the limited body of literature on the effects that the change in duties had on the well-being of frontline professionals fighting against COVID-19 [18,19]. Although evidence has been provided to support that healthcare professionals who had to change working wards during the pandemic were more likely to report psycho-physical symptoms [10,55], this is the first study to demonstrate that the change in professional duties during the COVID-19 pandemic was positively associated with moral distress and to identify the latter as a mediator in the relationship between the change in duties and psycho-physical malaise.

In doing so, this study also extends the findings of Miljeteig et al., by clarifying under which conditions healthcare workers in contact with COVID-19 patients may be at higher risk of experiencing moral distress [10]. Indeed, our results showed that healthcare professionals’ concern about being infected with the virus strengthened the positive association between the change in duties during the health emergency and total moral distress. In this regard, healthcare workers are at high risk of contracting the virus and transmitting it to their patients and families [17,56]. In the early stage of the pandemic, the uncertainty about the mode of transmission, including the infectivity of asymptomatic patients, working in frontline positions, the lack of appropriate PPE, and the initial lack of availability of any COVID-19 vaccine may have significantly impacted healthcare workers’ stress levels [56]. Interestingly, our results showed that the indirect effect of change in duties and concern about being infected through moral distress on psycho-physical malaise was significant for professionals who had high or moderate levels of concern about being infected with the virus at work, but not for those who had little concern about the risk of contagion. This is in line with a recent study [17] showing that healthcare professionals reported high levels of worry and fear of infection due to their work in close contact with COVID-19 patients and concern about getting sick and taking the infection home to family members. Indeed, working with an elevated risk of infection, insufficient PPE, heavy workloads, and human shortages may significantly increase moral distress.

Finally, our group comparisons based on socio-demographic variables revealed that women and physicians were more likely to experience psycho-physical malaise. Although the increased vulnerability of women to psycho-physical malaise has been well-established [57], research has found that physicians are generally more likely to enjoy good psycho-physical health than other healthcare professionals [58]. A possible explanation for this unexpected finding might be that although their greater decision-making authority and responsibilities may make physicians feel good in normal times, during the pandemic, they had to make tough decisions in the context of insufficient life-saving resources [13]. For instance, physicians had to make life or death triage decisions alone based solely on their clinical judgments [13]. The fact that physicians reported greater concern about contracting the virus than other professionals is in line with what was found in some previous studies showing that physicians perceived themselves to be at considerable risk of COVID-19 infection and a source of contagion to their loved ones, but they accepted this risk [59]. Additionally, unsurprisingly, younger healthcare professionals and those who had experienced the loss of a known person were more concerned about the COVID-19 contagion. Previous studies showed that younger professionals reported higher levels of anxiety and fear of COVID-19 contagion, probably because they had lower expertise in infectious disease management [60], and healthcare professionals who cared for COVID-19 patients who died reported greater worries and higher traumatic stress than those who cared for no COVID-19 deaths [61].

Although these findings are relevant in the current healthcare scenario, this study has some limitations. Firstly, because of the cross-sectional nature of our study, causal relationships cannot be inferred; therefore, a longitudinal design would be necessary to explore the causality of these relationships. Secondly, this research is based on the exclusive use of online self-reported measures and thus suffers the limitations of such a methodology. Future studies should collect data from multiple sources and utilize means to reach non-Internet users. Thirdly, our results might have been biased by the snowball sampling method. However, given its cost-effectiveness and efficiency in reaching difficult-to-track participants, snowball sampling has been largely used by prior studies that targeted healthcare professionals working during the COVID-19 outbreak [62,63]. Indeed, during the outbreak, it was complicated to collect data on the person due to physical distancing measures imposed by the government to contain the virus. Fourthly, this study presents only a first adaptation of the SCQ in the Italian context during the pandemic. Although our sample was mainly composed of women and limited to one Region located in Northern Italy, it included both nurses and physicians in direct contact with COVID-19 patients from different healthcare centers. This allowed us to provide an instrument having multi-professional applicability that may be helpful to practitioners in designing interventions targeting these professional groups. A future matter for further research would be to investigate whether the Italian version of the SCQ could also be suitable for other healthcare professional groups working in specific specialties. Replications are also needed in a post-pandemic setting to understand whether it may be a reliable instrument to assess moral distress during health emergencies. Further replications should also involve larger and more representative samples in terms of gender and region of origin (North, Centre, and South Italy) and deepen the psychometric properties of this instrument (e.g., measurement invariance based on socio-demographic characteristics). More research is needed to investigate whether the characteristics of the Italian adaptation of the SCQ could be transferable across diverse cultural contexts. It might be possible that although the original two-factor SCQ structure might be suitable for healthcare professionals working in Scandinavian settings, the unidimensional structure identified in this study and previously confirmed in its Australian validation might be suitable for healthcare professionals working in other European countries with a cultural configuration similar to the Italian one. In accordance with Glasberg et al., and åhlin et al., we call for more studies investigating the psychometric properties of the SCQ in other cultural settings [3,36]. Finally, given the lack of available agreed-upon definitions of low, moderate, and high levels of concern about the infection, we used arbitrary values defined by the distribution of the moderator in our specific sample (i.e., 16th, 50th, and 84th percentiles). Future research is needed to define cut-off values of concern about the infection with reference to European national populations and, accordingly, adopt them to better investigate the conditional effects of this variable.

Despite these limitations, this study provides practitioners with the first Italian version of the SCQ, a reliable instrument for evaluating moral distress among healthcare professionals working in current and future health emergencies. The findings of this study indicated that healthcare professionals who are called to perform tasks outside their usual clinical duties during emergencies are more likely to experience moral distress and then psycho-physical malaise. Thus, to limit the risk of moral distress, healthcare organizations should integrate emergency ethical preparedness into their management systems, establishing crisis response teams and introducing ethics frameworks that may guide their staff in difficult decision-making processes. Organizations could also provide their staff with “ethic rounds” and ethics training programs to prepare them to handle ethical demanding situations by strengthening their communication and ethical skills [64]. To maintain the psycho-physical well-being of their staff even in the aftermath of emergencies, hospitals should also introduce psychological support and counselling. Moreover, our findings suggest the importance of adequately training healthcare workers who are called to perform tasks outside their usual clinical duties during emergencies to prevent them from experiencing negative health outcomes. Indeed, this study indicated that these workers are more vulnerable to moral distress and then psycho-physical malaise when they are more (vs. less) concerned about the risk of contagion. Thus, as suggested by the differences found among professional groups that received appropriate versus inappropriate training and age groups, these workers should be trained and informed about how to perform these tasks safely in an emergency context (e.g., the use of PPE), and receive mentoring services to support them in performing unaccustomed tasks. To reduce their concern about the risk of infection, healthcare organizations should also promptly provide their staff working in emergencies with relevant information on guidelines/protocols designed to combat specific infectious diseases and training on infection control and disease management to increase their preparedness [65]. These training programs should also be integrated with training modules on critical thinking and clinical judgment, fundamental skills to improve healthcare professionals’ competence in infection control practice [65]. For instance, given that more (versus less) experienced healthcare professionals were at lower risk of moral distress and psycho-physical malaise, practitioners could organize training sessions and skill workshops in which senior healthcare workers, who were involved in prior disease outbreak episodes, are invited to share their experiences with younger colleagues. Additionally, human resource managers could consider including regular exercises in emergency prevention, preparedness, and response procedures in organizational health and safety management systems [66]. For instance, simulated drills of infectious disease outbreak scenarios that recreate a virtual experience of emergency response could be used to adequately prepare healthcare workers to handle morally distressing cases during future pandemic situations [65] and then maintain their well-being even during particularly challenging times.

## 5. Conclusions

This study enriches the literature on moral distress in two ways. It provides researchers and practitioners with the first Italian adaptation of the Stress of Conscience Questionnaire, a user-friendly reliable instrument for assessing moral distress among healthcare professionals during current and future public health emergencies. It is the first to identify moral distress as a psychological mechanism that explains why healthcare professionals who had to perform tasks outside their usual clinical duties during the COVID-19 pandemic were more likely to report psycho-physical malaise, especially when they were concerned about contracting the virus. We hope that healthcare organizations could learn from the lessons of the current pandemic by including regular ethical preparedness for health emergencies in their internal management systems, promptly preparing their staff, and then promoting their well-being.

## Figures and Tables

**Figure 1 ijerph-19-09682-f001:**
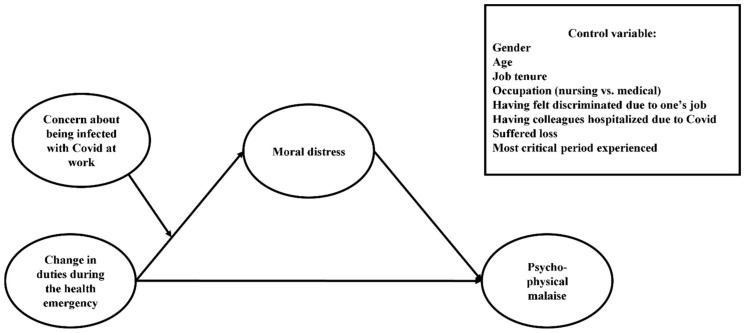
Conceptual model.

**Figure 2 ijerph-19-09682-f002:**
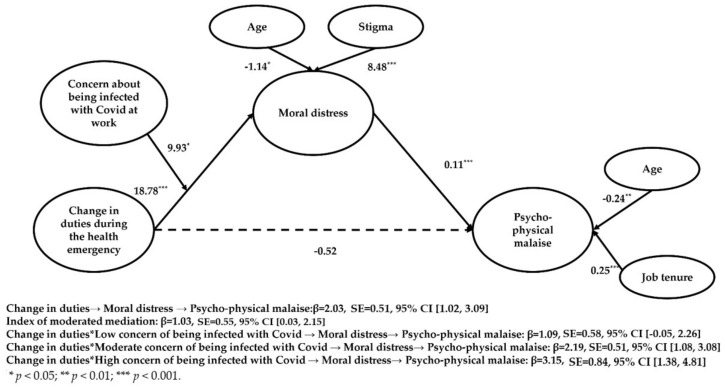
Path coefficients for the moderated mediation model.

**Figure 3 ijerph-19-09682-f003:**
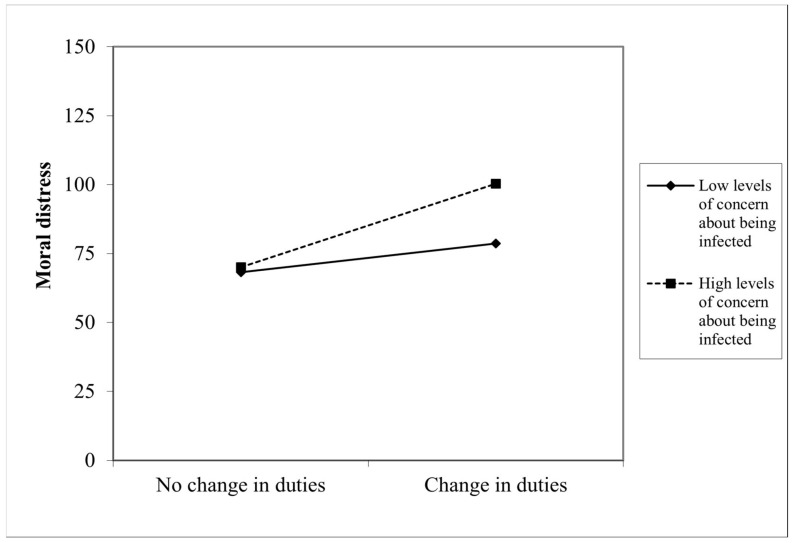
Moderating effect of concern about being infected with COVID during work.

**Table 1 ijerph-19-09682-t001:** Descriptive statistics of the sample (n = 266).

	% (n)
**Gender**	
Women	73.20 (199)
**Profession**	
Nursing profession	50.00 (133)
Medical profession	50.00 (133)
**Ward**	
Medical wards	21.60 (55)
Emergency rooms	18.00 (46)
Infectious disease wards	9.00 (23)
Service of anesthesiology and intensive care	8.60 (22)
Surgical departments	9.80 (25)
COVID-19 units	4.70 (12)
Intensive care units	3.10 (8)
Other wards	25.20
**Geographic location**	
Lombardy Region	92.6 (140)
Other regions in Northern Italy (Aosta Valley, Liguria, Emilia-Romagna, Piedmont, Veneto)	6.60 (18)
Regions in Central (Lazio, Tuscany, Umbria) and Southern Italy (Campania)	3.00 (8)
**Since the pandemic started, have you performed duties other than your usual ones?**	
Yes	50.00 (136)
**If yes, how has this change in duties been?**	
On an ongoing basis	54.90 (78)
On an occasional basis	45.10 (64)
**Have you received additional training for this change in duties?**	
Yes	45.40 (64)
**If yes, the training provided was:**	
poorly suited to your needs of performing a different role	25.30 (20)
useful for acquiring barely sufficient basic skills to function in emergencies	49.40 (39)
useful for acquiring necessary skills to function effectively in your different role	25.30 (20)
**Have any of your colleagues been diagnosed with COVID-19?**	
Yes, without hospitalization	47.60 (123)
Yes, with hospitalization	52.40 (142)
**Have any of your known persons (i.e., acquaintance, colleagues, families) been among fatalities?**	
Yes	53.00 (144)
**The most critical period in your work during the assistance of patients with COVID has been:**	
During the lockdown (February 2020–May 2020)	73.80 (200)
During a later stage of pandemic management (September 2020–February 2021)	19.60 (53)
The present period	6.60 (18)

**Table 2 ijerph-19-09682-t002:** Content validity indices regarding the three expert opinions.

Item	Q1: to What Extent Do You Agree That the Item Is Representative of the Definition of Moral Distress?	Q2: to What Extent This Item Is Essential to the Measure of the Construct of Moral Distress?	Expert in Agreement on the Understandability of the Item	Expert in Agreement on Keeping the Item	I-CVI
Item 1	5.00	3.00	3	3	1
Item 2	5.00	3.00	3	3	1
Item 3	5.00	3.00	3	3	1
Item 4	5.00	3.00	3	3	1
Item 5	4.33	2.67	3	3	1
Item 6	3.33	2.33	3	3	1
Item 7	4.00	2.67	3	3	1
Item 8	4.00	2.67	3	3	1
Item 9	3.33	2.67	3	3	1
Scale of response	1 (Totally disagree)– 5 (Totally agree)	1 (Useless)–3 (Essential)		**S-CVI/Ave**	1
				**S-CVI/UA**	

Note. I-CVI: item-level content validity index; UA: Universal agreement; S-CVI/Ave: scale-level content validity index based on the average method; S-CVI/UA: scale-level content validity index based on the universal agreement method [40].

**Table 3 ijerph-19-09682-t003:** Descriptive statistics of Stress of Conscience Questionnaire items.

	Descriptive Statistics				Factor Loading
Scale Item	M	SD	Skewness	Kurtosis	1
Item 1	12.29	8.34	0.05	−1.19	0.77
Item 2	6.24	7.53	1.13	0.10	0.75
Item 3	9.07	8.25	0.53	−0.90	0.77
Item 7	13.87	9.18	−0.16	−1.37	0.74
Item 8	8.56	8.69	0.65	−0.96	0.75
Item 9	5.32	7.41	1.41	1.02	0.53
Explained variance (%)	52.08%				
Cronbach’s alpha	0.81				
McDonald’s omega	0.82				

Note. *h^2^* = *item communality*. *Factor loadings > |0.40| are in bold.* Explained variance and McDonald’s omega refer to the one-factor solution including the selected six items.

**Table 4 ijerph-19-09682-t004:** Intercorrelations among study variables, socio-demographic, and COVID-19-related experience variables (n = 266).

	M	SD			Skew.	Kurt.	1	2	3	4	5	6	7	8	9	10
1. Gender	-	-			-	-	-									
2. Age	45.26	11.57			-	-	−0.05 ^a^	-								
3. Job tenure	17.81	11.46			-	-	−0.03 ^a^	0.89 **^,a^	-							
4. Profession	-	-			-	-	0.14 *^,b^	−0.28 ***^,a^	−0.08 ^a^	-						
5. Having hospitalized colleagues	-	-			-	-	−0.05 ^b^	0.06 ^a^	0.08 ^a^	0.02 ^c^	-					
6. Loss of someone due to COVID	-	-			-	-	0.13 *^,b^	0.05 ^a^	0.07 ^a^	0.06 ^c^	0.17 **^,c^	-				
7. Most critical time experienced	-	-			-	-	0.02 ^b^	0.18 **^,a^	0.15 *^,a^	−0.06 ^c^	0.11 ^c^	0.11 ^c^	-			
8. Change of duties							−0.06 ^b^	0.09 ^a^	0.05 ^a^	−0.26 ***^,c^	0.19 **^,c^	0.03 ^c^	0.07 ^c^	-		
9. Moral distress	55.36	35.62			0.37	−0.50	0.07 ^a^	−0.13 **^,c^	−0.09 *^,c^	0.01 ^a^	0.01 ^a^	0.08 ^a^	0.02 ^a^	0.14 **^,a^	**0.81**	
10. Psycho-physical malaise	18.62	6.91			0.15	−0.37	0.15 *^,a^	−0.11*^,c^	−0.02 ^c^	0.14 *^,a^	0.03 ^a^	0.07 ^a^	−0.06 ^a^	−0.03 ^a^	0.60 **^,d^	**0.82**
	**MED**	**IQR**	**Q_1_**	**Q_3_**	**Skew.**	**Kurt.**	**1**	**2**	**3**	**4**	**5**	**6**	**7**	**8**	**9**	**10**
11. Concern about the infection	3.00	1.00	2.00	3.00	−0.35	−0.31	0.02 ^a^	−0.06 ^a^	−0.01 ^a^	0.20 **^,a^	0.06 ^a^	0.15 *^,a^	−0.08 ^a^	0.28 ^c^	0.17 **^,a^	0.12 *^,a^
12. Having felt discriminated	1.68	1.00	1.00	2.00	0.70	−0.89	0.19 **^,a^	−0.19 **^,a^	−0.09 ^a^	0.18 **^,a^	−0.07 ^a^	0.09 ^a^	−0.14 *^,a^	−0.09 ^c^	0.26 **^,a^	0.27 **^,a^

Note. Boldfaced numbers on the diagonal represent Cronbach’s alpha; M = means; SD = standard deviations; Skew. = Skewness; Kurt. = Kurtosis; MED = medians; IQR = interquartile range; Q_1 =_ first quartile or 25th percentile; Q_3_ = third quartile or 75th percentile. * *p* < 0.05; ** *p* < 0.01; *** *p* < 0.001; ^a^ Spearman’s rho correlation coefficient; ^b^ Phi coefficient obtained from contingency table; ^c^ Kendall’s coefficient of rank correlation tau-sub-b coefficient, ^d^ Pearson’s correlation coefficient. Gender: 0 = men, 1 = women; Age = in years; Job tenure: in years; Profession: 0 = nursing profession, 1 = medical profession; Hospitalized colleagues: 0 = no, 1 = yes; Social stigma: from 1 = never, 4 = often; Loss of someone: 0 = no, 1 = yes; Most critical time: 0 = others, 1 = lockdown.

**Table 5 ijerph-19-09682-t005:** Results of the independent *t*-test and Mann-Whitney test in relation to gender, profession, modalities of change of duties, receiving supplementary training for change in duties, having hospitalized colleagues, having suffered a loss.

	Men (n = 73)					Women (n = 199)					t	*p*	95% CI
	M	SD				M		SD					LL
Moral distress	51.03	31.67				46.90		36.90			−1.22	0.225	−15.51
Malaise	16.97	6.37				19.23		7.02			−2.41	0.017	−4.10
	MED	IQR	Q_1_	Q_2_	MR	MED		IQR	Q_1_	Q_2_	MR	z	*p*
Concern	3.00	1.00	2.00	3.00	129.49	3.00		1.00	2.00	3.00	132.89	−0.34	0.731
	**Nurses** **(n = 133)**					**Doctors** **(n = 133)**					**t**	** *p* **	**95% CI**
	**M**	**SD**				**M**		**SD**					**LL**
Moral distress	54.85	32.84				56.07		37.50			−0.28	0.777	−9.73
Malaise	17.62	7.03				19.62		6.37			−2.44	0.015	−3.63
	MED	IQR	Q_1_	Q_2_	MR	MED		IQR	Q_1_	Q_2_	MR	z	*p*
Concern	3.00	1.00	2.00	3.00	142.96	3.00		0.00	3.00	3.00	115.62	−3.17	0.002
	**Change of duties on an ongoing basis** **(n = 64)**					**Change of duties on an occasional basis** **(n = 78)**					**t**	** *p* **	**95% CI**
	**M**	**SD**				**M**		**SD**					**LL**
Moral distress	59.23	33.96				66.78		36.44			−1.27	0.208	−19.33
Malaise	19.19	7.41				18.88		5.95			0.27	0.787	−1.91
	MED	IQR	Q_1_	Q_2_	MR	MED		IQR	Q_1_	Q_2_	MR	z	*p*
Concern	3.00	1.75	2.25	4.00	56.98	3.00		1.00	2.00	3.00	77.60	−3.22	0.001
	**No additional training** **(n = 77)**					**Additional training** **(n = 64)**					**t**	** *p* **	**95% CI**
	**M**	**SD**				**M**		**SD**					**LL**
Moral distress	60.70	38.05				65.25		32.95			−0.75	0.454	−16.53
Malaise	19.06	6.87				18.73		6.42			0.29	0.770	−1.91
	MED	IQR	Q_1_	Q_2_	MR	MED		IQR	Q_1_	Q_2_	MR	z	*p*
Concern	3.00	1.00	2.00	3.00	60.85	3.00		2.00	2.00	4.00	77.21	−2.56	0.010
	**Non-hospitalized colleagues** **(n = 129)**					**Hospitalized colleagues** **(n = 142)**					**t**	** *p* **	**95% CI**
	**M**	**SD**				**M**		**SD**					**LL**
Moral distress	55.07	35.13				56.01		35.99			−0.22	0.827	−9.46
Malaise	18.32	7.20				18.99		6.58			−0.79	0.426	−2.31
	MED	IQR	Q_1_	Q_2_	MR	MED		IQR	Q_1_	Q_2_	MR	z	*p*
Concern	3.00	1.00	2.00	3.00	127.02	3.00		1.00	2.00	3.00	135.40	−0.96	0.335
	**No loss** **(n = 197)**					**Suffered loss** **(n = 113)**					**t**	** *p* **	**95% CI**
	**M**	**SD**				**M**		**SD**					**LL**
Moral distress	52.53	36.25				57.87		34.98			−1.24	0.217	−13.85
Malaise	18.11	7.56				19.08		6.27			−1.16	0.247	−2.63
	MED	IQR	Q_1_	Q_2_	MR	MED		IQR	Q_1_	Q_2_	MR	z	*p*
Concern	3.00	1.00	2.00	3.00	120.79	3.00		2.00	2.00	4.00	142.16	−2.46	0.014

Note. M = means; SD = standard deviations; t = *t*-value; *p* = *p*-value; 95% CI = 95% confidence interval; MED = medians; IQR = interquartile range; Q_1_ = first quartile or 25th percentile; Q_3_ = third quartile or 75th percentile; MR = mean range; z = z value.

**Table 6 ijerph-19-09682-t006:** ANOVAs and Kruskal-Wallis H tests between age, years of experience, quality of training, most critical period experienced groups regarding the study’s variables.

Dimension	Job Tenure	*Mean*	*SD*		*F*		
**Moral distress**	≤10 years	59.49	37.91		1.33		
	11–20 years	56.25	29.90				
	>20 years	51.22	37.42				
**Psycho-physical malaise**	≤10 years	18.53	6.58		0.50		
	11–20 years	18.09	6.95				
	>20 years	19.10	7.19				
		** *MED* **	** *IQR* **	**Q_1_**	**Q_2_**	** *MR* **	** *X^2^* **
**Concern about the infection**	≤10 years	3.00	1.00	2.00	3.00	136.28	0.59
	11–20 years	3.00	1.00	2.00	3.00	131.98	
	>20 years	3.00	1.00	2.00	3.00	128.37	
**Dimension**	**Age**	** *Mean* **	** *SD* **	** *F* **			
**Moral distress**	≤35 years	58.23	38.27	5.28 **			
	36–45 years	63.62	31.51				
	46–55 years	57.57	33.65				
	>55 years	38.73	35.38				
**Psycho-physical malaise**	≤35 years	19.20	6.46	2.09			
	36–45 years	19.44	6.81				
	46–55 years	18.88	7.01				
	>55 years	16.52	7.21				
		** *MED* **	** *IQR* **	**Q_1_**	**Q_2_**	** *MR* **	** *X^2^* **
**Concern about the infection**	≤35 years	3.00	1.00	3.00	4.00	152.38	19.92 ***
	36–45 years	2.00	1.00	2.00	3.00	96.66	
	46–55 years	3.00	1.00	2.00	3.00	136.20	
	>55 years	3.00	1.00	2.00	3.00	135.91	
**Dimension**	**Most critical period experienced**	** *Mean* **	** *SD* **		** *F* **		
**Moral distress**	Lockdown	56.10	35.62		0.16		
	Later stage	54.45	37.01				
	Present period	51.55	33.12				
**Psycho-physical malaise**	Lockdown	18.45	6.67		2.79		
	Later stage	20.24	6.87				
	Present period	16.05	8.87				
		** *MED* **	** *IQR* **	**Q_1_**	**Q_2_**	** *MR* **	** *X^2^* **
**Concern about the infection**	Lockdown	3.00	1.00	2.00	3.00	128.20	6.06
	Later stage	3.00	1.00	2.00	3.00	129.53	
	Present period	3.00	1.00	2.00	3.00	111.27	
**Dimension**	**Quality of training received**	** *Mean* **	** *SD* **		** *F* **		
**Moral distress**	Inadequate	80.75	29.22		6.87 **		
	Basic	63.87	33.79				
	Adequate	42.35	34.44				
**Psycho-physical malaise**	Inadequate	18.10	4.30		1.22		
	Basic	19.10	6.21				
	Adequate	16.40	7.93				
		** *MED* **	** *IQR* **	**Q_1_**	**Q_2_**	** *MR* **	** *X^2^* **
**Concern about the infection**	Inadequate	4.00	0.25	3.75	4.00	51.39	12.56 **
	Basic	3.00	1.75	2.00	3.75	33.35	
	Adequate	3.00	1.00	2.00	3.00	30.47	

Note. M= means; SD= standard deviations; F = F value; MED = medians; IQR = interquartile range; Q_1=_ first quartile or 25th percentile; Q_3=_ third quartile or 75th percentile; MR = mean range; X^2^ = chi-square value. ** *p* < 0.01; *** *p* < 0.001.

**Table 7 ijerph-19-09682-t007:** Path coefficients and direct and conditional effects for the moderated mediation model.

Model (Outcome)	R	R^2^	*p*	MSE	F	df_1_
*Model 3a (Moral distress)*	0.47	0.33	0.00	998.25	6.20	11
*Model 3b (Psycho-physical malaise)*	0.63	0.40	0.00	26.87	16.29	10
**Standardized direct and conditional effects**						
** *Effects—Model 3a* **		*B*		*SE*	t	*p*
Change in duties → Moral distress		18.78 ***		4.21	4.53	0.00
Concern → Moral distress		5.84 *		2.58	2.26	0.02
Change in duties * Concern → Moral distress		9.93 *		5.01	1.98	0.04
Gender → Moral distress		1.62		4.67	0.35	0.73
Age → Moral distress		−1.14 *		0.47	−2.43	0.01
Job tenure → Moral distress		0.56		0.45	1.25	0.21
Occupation → Moral distress		−4.84		4.94	−0.98	0.33
Stigma → Moral distress		8.48 ***		2.47	3.43	0.00
Hospitalized colleagues → Moral distress		−0.96		4.15	−0.23	0.82
Most critical period experienced → Moral distress		4.55		4.49	1.01	0.31
Suffered lost → Moral distress		3.78		4.20	0.90	0.37
** *Effects—Model 3b* **		*B*		*SE*	t	*p*
Change in duties → Malaise		−0.52		0.72	−0.73	0.47
Stress → Malaise		0.11 ***		0.01	10.07	0.00
Gender → Malaise		1.01		0.76	1.32	0.19
Age → Malaise		−0.24 **		0.08	−3.13	0.00
Job tenure → Malaise		0.25 ***		0.07	3.42	0.00
Occupation → Malaise		0.09		0.80	0.11	0.91
Having felt discriminated → Malaise		0.35		0.41	0.85	0.39
Having hospitalized colleagues → Malaise		0.45		0.68	0.67	0.50
Most critical period experienced → Malaise		−0.67		0.73	−0.91	0.36
Suffered lost → Malaise		0.04		0.68	0.06	0.95
** *Conditional effects* **		*B*			*SE*	
Change in duties * Low Concern → Distress →Malaise		1.09			0.58	
Change in duties * Moderate Concern→ Distress →Malaise		2.19			0.51	
Change in duties * High Concern→ Distress →Malaise		3.15			0.84	

Note. Low, moderate, and high values of concern about the infection refer to the 16th (i.e., 2), 50th (i.e., 3), and 84th (i.e., 4) percentiles. * *p* < 0.05; ** *p* < 0.01; *** *p* < 0.001.

## Data Availability

Data are available from the authors upon reasonable request.

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
