# Peer review of "Moral Distress among Frontline Physicians and Nurses in the Early Phase of COVID-19 Pandemic in Italy"

_ijerph, 2022, doi:10.3390/ijerph19159682_

Round 1
Reviewer 1 Report
Thank you for the opportunity to read this Manuscript. Moral distress is a global concern among healthcare professionals. For this reason, I consider the manuscript “Moral distress among frontline physicians and nurses in the early phase of COVID-19 pandemic in Italy” as innovative and important, especially to those with closely related topics. The purpose of the paper is well defined, the methods are appropriate, the data is sound and overall the manuscript adheres to the relevant standards for reporting psychometric properties. However, I have a few concerns and suggestions for major and minor revisions.
· INTRODUCTION: I am not sure that literature considers “moral distress” and “stress of conscience” as synonyms. Could you provide a citation about that?
· INTRODUCTION: When you are talking about the instruments to measure moral distress, you show only two instruments. However, several literature reviews show that there are a lot of instruments such as the MDS-revised, the moral distress risk scale (MDSR), the moral distress thermometer (MDT) and so on. Based on that, why do you focus your attention only on MDS and stress of conscience questionnaire? Please, let us know the rationale.
· SAMPLE: please, put all the descriptive statistics of the sample in the result section.
· METHODS: How did authors actually recruit participants? More details are needed.
· METHODS: How did authors collect data for psychometric analysis, e.g. sample selection, who collected the data, social networks or mailing lst, etc., and please explain?
· METHODS: Did you send out reminders to increase the response rates? If yes, please give a brief description.
· The value of the article could also be increased by extending the Discussion section with a discussion of the practical Implications of the results and development of the literature implemented in the introduction.
· Limitations: To what extent are the scale characteristics not transferable across translation?
· Given the limitations presented, what are the potentials to minimize the limitations described?
Reviewer 2 Report
A well-written work. This is a methodological study, not a cross-sectional one. For the items in the scale, information should be given about the content validity rate and content validity indexes regarding the expert opinions.
Reviewer 3 Report
Dear Authors,
The manuscript on the important and timely issue. The Introduction section is well written and provided relevant and up-to-date background for the research. You should decide what the main issue of the manuscript is: to establish psychometric properties of the Italian version of the SCQ or investigate distress among health-care providers. In the validation study, you should demonstrate that even though significant modification was done for the tool, still, it is relevant to apprise moral distress in local context. In the deductive study, which can be done only after the first one had been completed, you can talk about the complex phenomena of interplay between distress, psycho-physical health and pandemic impact. The study which primary aims are establish validity of the tool in a new context (language, state, population), may be hardly used to draw conclusions on phenomena, but rather should be based on well-known and established phenomena. The statistical analysis of some variables of the study did not adequate for their measurement, limiting the interpretation of the findings. My detailed comments are attached below.
Introduction:
Lines 75-78: You repeat the idea from the previous sentence. Please revise.
Lines 95-98: Again, you come back to the same issue which was addressed in the previous sentence. I recommend to merge them.
Most importantly, the relevance of the second study aim is not clear given a general topic of the SCQ validation.
Materials and Methods:
Line 154: The report on the ethic procedures should be done in accordance to the ethical standards. Written informed concern does not inclusion criteria, but prerequisite to participate in the study.
In addition, please detail exclusion criteria for the study participation.
The SCQ:
Lines 184-195: The SCQ is a main tool in the study, so it should be describe in a clear and consolidated way for understanding of the results. To this aim,
1. please move to this paragraph the description of the translation procedures to Italian (Lines 225-240).
2. Provide details if other types of adaptations were done to the local context (as you reported in lines 21, 133, etc.,) besides removing of 4 items from 10. And,
3. importantly, provide detailed description of the psychometric properties of the original tool.
4. If you want to use the tool to investigate distress impact, you should report on psychometric properties of the tool. Based on your description, I understood that you use only 6 items from the original scale for the analysis (moral distress variable in the study). Given considerable modification of the tool, you should provide information on basic reliability and extensive validity of the reduced tool. Otherwise, you can’t use the tool for the phenomena investigation. All the results with this tool are under the question if it is a real situation or problems of the way to measure the distress…
The “Concern about the infection” was measured with one single item, which was rated on 4=point Likert scale. Such type of measure does not suitable for analysis with tests for normal distribution. The data should be re-analyzed with non-parametric tests for both correlations and differences. In addition, you should to describe such a data with median and inter-quadrantile range instead of mean and SD. The same is right for all the analyses (correlation, ANOVA, reports like this: “Healthcare providers who received additional training reported greater concern (M=3.07, SD=.85) than those who did not received it (M=2.67, SD=.82) (Line 306-307), etc.)
In addition, it is not clear what is your definition for “low” and “high” concern based on the scoring as you present in the moderated mediation analysis.
Line 193:” …which is computed by summing the products…” Do you mean “scores” or “ratings” under “products”?
Results
Table 3: It is not clear which type of correlation tests were used for each pair of variables. For example, if the correlation between job tenure and age was investigated with the same test as correlation between “having hospitalized colleagues” (yes/no) and gender? The last should be investigated with χ2. The reported values does not fit the significance threshold for significance (see pair 1 and 9 variables).
Discussion
Line 389 – “Thus, it might be that the three excluded items” – your report is confusing. In the Methods section you informed that 4 items were removed.
The focus of the discussion should be consistent with the study aims. Even though you report that the findings are consistent with the hypothesis supporting the tool validity (Line 409), your next discussion deals with presentation of “newly revealed” trends (Line 415).
Line 428 and so on – Again, the relevance of this issue to the general manuscript topic is not clear.
There are too many references in the manuscript. Please cite only the most relevant and updated ones.
Round 2
Reviewer 1 Report
Good job! Congratulation.
Author Response
Thank you very much for appreciating the efforts made to review our manuscript. We proofread the manuscript and checked the English language and style using a grammar editing software (i.e., Grammarly). We would like to thank you again for your constructive comments that helped us improve the quality of our manuscript.
Reviewer 3 Report
Thank you very much for addressing my major concerns. The current version of the manuscript provides the required information for understanding the research and its results. Minor comment: IQR is a disparity measure and should be reported using two numbers: 25 percentile and 75 percentile.
Good luck.
Author Response
(Legend: R3.1 stands for Reviewer 3, comment 1, and A3.1 for our reply to this comment)
RESPONSES TO REVIEWER #3’S COMMENTS
Thank you very much for addressing my major concerns. The current version of the manuscript provides the required information for understanding the research and its results.
Thank you very much for appreciating our revised version of the manuscript.
R3.1
Minor comment: IQR is a disparity measure and should be reported using two numbers: 25 percentile and 75 percentile. Good luck.
A3.1
Thank you for this important note. Accordingly, we have added the information about the first and third quartiles (i.e., 25th and 75th percentiles) in both the text and tables, clarifying it in our revised notes of the tables. We would like to thank you again for your insightful comments that helped us improve the quality of our manuscript.